# Effect of Salinity on Stomatal Conductance, Leaf Hydraulic Conductance, HvPIP2 Aquaporin, and Abscisic Acid Abundance in Barley Leaf Cells

**DOI:** 10.3390/ijms232214282

**Published:** 2022-11-18

**Authors:** Guzel Sharipova, Ruslan Ivanov, Dmitriy Veselov, Guzel Akhiyarova, Oksana Seldimirova, Ilshat Galin, Wieland Fricke, Lidiya Vysotskaya, Guzel Kudoyarova

**Affiliations:** 1Ufa Institute of Biology of Ufa Federal Research Centre of the Russian Academy of Sciences, pr. Octyabrya 69, 450054 Ufa, Russia; 2School of Biology and Environmental Science, University College Dublin, Belfield, D04 V1W8 Dublin, Ireland

**Keywords:** aquaporin, HvPIP2, salinity, hydraulic conductance, abscisic acid

## Abstract

The stomatal closure of salt-stressed plants reduces transpiration bringing about the maintenance of plant tissue hydration. The aim of this work was to test for any involvement of aquaporins (AQPs) in stomatal closure under salinity. The changes in the level of aquaporins in the cells were detected with the help of an immunohistochemical technique using antibodies against HvPIP2;2. In parallel, leaf sections were stained for abscisic acid (ABA). The effects of salinity were compared to those of exogenously applied ABA on leaf HvPIP2;2 levels and the stomatal and leaf hydraulic conductance of barley plants. Salinity reduced the abundance of HvPIP2;2 in the cells of the mestome sheath due to it being the more likely hydraulic barrier due to the deposition of lignin, accompanied by a decline in the hydraulic conductivity, transpiration, and ABA accumulation. The effects of exogenous ABA differed from those of salinity. This hormone decreased transpiration but increased the shoot hydraulic conductivity and PIP2;2 abundance. The difference in the action of the exogenous hormone and salinity may be related to the difference in the ABA distribution between leaf cells, with the hormone accumulating mainly in the mesophyll of salt-stressed plants and in the cells of the bundle sheaths of ABA-treated plants. The obtained results suggest the following succession of events: salinity decreases water flow into the shoots due to the decreased abundance of PIP2;2 and hydraulic conductance, while the decline in leaf hydration leads to the production of ABA in the leaves and stomatal closure.

## 1. Introduction

The balancing of water losses and uptake by plants enables the maintenance of hydraulic homeostasis. Stomatal movement is an important mechanism preventing potentially harmful fluctuations in transpiration-induced changes in leaf water potential and turgor. Under conditions of lowered availability of water, stomatal closure reduces transpiration, which helps to maintain the hydration of tissues. The main attention of researchers has been addressed to the active uptake and efflux of osmotically active solutes into and out of stomatal guard cells as the mechanism controlling stomatal movement [1]. However, it was shown that stomatal closure may be controlled by a decline in water flow into the leaves due to the reduced ability of leaf tissues to conduct water [2]. Nevertheless, although a number of works show a functional relationship between the hydraulic and stomatal conductivities of the leaf [3], the mechanisms underlying this relationship remain insufficiently studied. The stomatal conductance is expected to coordinate with the leaf hydraulic conductance [4]. However, empirical information about their coordination is scarce. It is stated that there is currently no general opinion about the actual contribution and relative importance of stomatal closure driven by active (ABA-mediated) and/or passive (hydraulic-mediated) mechanisms in modulating stomatal closure in planta [5]. Water conductance by plant tissues depends on the presence and activity of water channel aquaporins in the membranes of cells [6]. Although transpiration and aquaporins are considered as main components influencing plant water status, their relations have been investigated in detail only recently and still remains insufficiently studied [7]. The aim of the present work was to study the relationship of hydraulic and stomatal conductance in salt-stressed plants and to test for any involvement of aquaporins (AQPs) in stomatal closure under salinity. Previously, we have shown rapid stomatal closure after the addition of sodium chloride to the nutrient medium of barley plants [8], associated with the fast accumulation of abscisic acid (ABA) [8], the hormone known to be involved in the control of stomatal closure [9]. ABA is also known to influence hydraulic conductance by increasing the activity of AQPs [6]. The increased abundance of AQPs has been detected in barley roots treated with exogenous ABA [10]. However, the relationship between stomatal conductance and shoot hydraulic conductivity and their possible relationship with the changes in ABA concentration were not considered in the salt-stressed barley plants. In the present work, the changes in the level of aquaporins in the leaf cells of salt-stressed barley plants were detected with the help of an immunohistochemical technique using antibodies against AQPs, as described previously [10]. In parallel, leaf sections were stained for ABA. The changes in ABA content in the leaves were related to the influx of the hormone from the roots to reveal the role of the ABA coming from the roots in the regulation of stomatal and hydraulic conductance in leaves. Many researchers have argued for the importance of root-derived ABA in the control of stomatal conductance [11,12]. However, this notion was not supported by other reports showing stomatal closure induced by ABA synthesized in the leaves themselves [13]. The present work was undertaken to clarify if the stomata of salt-stressed barley roots were closed under the effect of ABA transported from roots or synthesized in the leaves themselves. Furthermore, the effects of salinity were compared to those of exogenously applied ABA on leaf AQP levels and the stomatal and leaf hydraulic conductance of barley plants. The results obtained by us suggest the following succession of events: in the short term, salinity decreases water flow into the shoots due to the decreased abundance of PIP2;2 bringing about reduced hydraulic conductance, while the decline in leaf hydration leads to the production of ABA in the leaves and stomatal closure.

## 2. Results

An analysis of the effect of salinity on the transpiration of intact plants showed a 20% decline in water loss as compared with the control already 20 min after addition of NaCl to the nutrient solution of barley plants (Appendix A).

Measurement of the transpiration of detached leaves showed a decrease in transpiration in response to NaCl and ABA (Figure 1A). Meanwhile, the leaf water potential was lowered by salinity by about 0.18 MPa but was not affected by the ABA treatment (Figure 1B). Calculation of the hydraulic conductivity showed that it was significantly decreased by salinity and increased by the ABA treatment (Figure 1C). The addition of ABA to salt-treated plants resulted in about a 1.5 times increase in hydraulic conductance compared to the plants untreated with ABA and grown under salinity.

It was important to find out whether the changes in hydraulic conductance caused by the addition of NaCl were related to the abundance of AQPs in leaves. To answer this question, we performed an immunolocalization of AQPs in the leaves of barley with the help of specific antibodies against HvPIP2;2. This PIP2 isoform had been found in preliminary experiments to show the largest changes in abundance compared with other PIP2 antibodies. Figure 2A shows intensive staining for HvPIP2;2 at the periphery of cells of the vasculature and especially of the mestome sheath cells. The addition of sodium chloride to the nutrient medium weakened the staining for HvPIP2;2 of the leaf cells (Figure 2). The effect was most clearly manifested at the site of the vasculature. An immunochemical control based on the use of an unspecific serum shows low (almost absent) section staining and confirms the specificity of immunolocalization using specific antibodies.

The opposite effect was detected in the leaves of ABA-treated plants (Figure 3C), where an increase in staining for PIP2;2 was revealed. The averaged results of a semiquantitative estimation of the staining for PIP2;2 of the vasculature cells with the help of the ImageJ program confirmed its decrease in the vasculature of the salt-treated plants and its increase in the case of ABA treatment (Figure 3D). The changes in the PIP2;2 abundance were in agreement with those in hydraulic conductance induced by NaCl and ABA treatment (Figure 1C): the PIP2;2 abundance and hydraulic conductance declined in the case of salt-treated plants and increased in both characteristics in ABA-treated plants compared to the control.

It was of interest to estimate the level of ABA in the cells under salinity. Figure 4 shows that the intensification of the staining for ABA by NaCl is best noticeable in the chloroplasts along the periphery of the mesophyll cells and stomata. In the ABA-treated plants, the increased staining for this hormone was detected mostly in the vasculature. A high level of ABA was detected with immunostaining in chloroplast-like structures, which is in accordance with the “anion trap” concept [14]. As a weak acid, ABA is mostly uncharged when present in the relatively acidic compartments of plants and can easily diffuse across membranes, while it dissociates and accumulates in an alkaline compartment like the stroma of illuminated chloroplasts.

To detect the formation of apoplast barriers, we used berberine staining of lignin and suberin. Their staining showed increased fluorescence in the periphery of the cells of the vasculature (Figure 5), coinciding with increased staining for PIP2;2 (Figure 2).

Salt treatment decreased the turgor of the ridge cells overlying large lateral veins but had no effect on their hydraulic conductivity (Table 1). The osmolality of the cell sap averaged about 550 mosmol kg^−1^ and remained unchanged during this short-term (20 min) exposure to salinity (not shown).

The ABA concentration in the xylem sap of plants averaged about 20 ng mL^−1^ and was not increased by salinity (21 ± 2 and 20 ± 3 ng mL^−1^ in the xylem of the control and salt-treated plants, respectively).

## 3. Discussion

Similar to the previous experiments [8,15], a decline in transpiration was detected shortly after the start of the salinity treatment. This response was due to lower leaf stomatal conductance, which, as shown earlier, decreased by about 2 times in half an hour after the start of the salt treatment [8]. The appearance of NaCl ions in the nutrient solution reduces the ability of the plants for water uptake, while the decline in stomatal conductance is an adaptive response decreasing transpirational water losses, thereby protecting plants from dehydration. The goal of the present work was to find out whether the decline in transpiration was related to the leaf hydraulic conductance.

Calculation of leaf hydraulic conductance showed that it was decreased by salinity. The decrease in leaf hydraulic conductance was accompanied by a lower (more negative) leaf water potential and a decrease in cell turgor. Thus, the reduced hydraulic conductance was likely to contribute to the decline in leaf hydration resulting from the salt treatment. This effect coincided with the decline in PIP2;2 in the vasculature. A high level of staining for PIP2;2 was detected in the cells of the vascular bundles of control plants, indicating its great abundance. These results are in accordance with the information about increased activity of PIP2;2 at the sites of the most intensive water flow across cell membranes. Since water is delivered to the leaves through the xylem vessels, the presence of AQPs in the membranes of the bundle sheath is not surprising. The staining was greater in the mestome sheath due to it being the more likely hydraulic barrier due to the deposition of lignin [16]. The present experiments confirmed the lignification of the cells of the mestome sheath (Figure 5), and the formation of barriers for water movement is known to increase the importance of AQPs for the control of hydraulic conductivity, as AQPs allow water to move from cell to cell rather than along the apoplast, where barriers such as lignin are deposited. The importance of the decline in PIP2;2 abundance in the mestome sheath of salt-stressed plants was confirmed by its correlation with the salt-induced decline in leaf hydraulic conductance, unlike the hydraulic conductivity of ridge cells overlying large lateral veins, which remained unchanged by salinity.

Pantin and coauthors [2] suggested that the decline in the hydraulic conductivity of the water pathway from the vessels through the mesophyll may contribute to stomatal closure. The possible impact of AQPs was only suggested, not experimentally tested, in that study. The results obtained in the present experiments show that the decline in the level of AQPs in the leaves in response to salinity may contribute to stomatal closure.

The increased immunostaining for ABA by the salt treatment is in accordance with previous data [17]. ABA is known to influence root hydraulic conductivity (e.g., [10]), while the data on its effect on leaf hydraulic conductance are scarce, rather contradictory, and obtained mainly on dicots. Consequently, we treated barley plants with ABA to find out whether an increased concentration of this hormone may influence leaf hydraulic conductivity. Figure 1C shows an ABA-induced increase in leaf hydraulic conductance paralleled with the increased abundance of AQPs (Figure 3C). The addition of ABA to salt-stressed plants increased hydraulic conductance, but the increase was still lower than in plants treated with ABA alone, suggesting that the NaCl and exogenous ABA acted in different ways.

It is important to note that the effect of exogenous ABA on the level of aquaporins was directly opposite to the changes in their level that were registered in the leaf during the salt treatment. While the ABA increased the leaf hydraulic conductance and AQP abundance of the vascular bundle sheath, salinity reduced both. Thus, the study of the effects of exogenous ABA does not support an assumption that the decrease in the level of aquaporins and hydraulic conductivity during salt treatment may be a consequence of the accumulation of ABA. The difference in the action of exogenous ABA and the accumulation of the endogenous hormone in salt-treated plants may be due to a difference in the distribution of ABA between cells induced by the two treatments. The salt-induced changes in the staining for ABA were less noticeable in the vascular bundles than in the mesophyll (Figure 4).

It is known [18] that the increase in ABA content in the leaves may be due to its elevated synthesis either in the roots or in the leaves themselves. The greater staining for ABA in mesophyll cells compared to the cells of the vascular bundles suggests that, under the present conditions, the accumulation of ABA in the leaves of salt-treated plants was more likely to be due to its synthesis in the leaves themselves. This assumption is supported by the results of measuring the ABA concentration in the xylem sap showing the absence of an increase in ABA concentration in the salt-stressed plants [19]. Accumulation of ABA is known to be stimulated by a decline in cell hydration [20]. A decline in leaf cell turgor detected in the present experiments (Table 1) suggests that decreased leaf hydration could serve as a stimulus for ABA accumulation. It was shown previously that salinity led to accumulation of osmolytes, resulting in turgor recovery and maintenance detected 20 h following addition of 100 mM NaCl to the root medium [21]. However, the action of the NaCl was too short (20–25 min) in the present experiments for osmotic adjustment to happen. The decrease in the level of AQPs detected in the present experiments in salt-treated barley plants was likely to lead to a decrease in the water flow to the leaf cells, bringing about the increase in ABA synthesis in the mesophyll. Thus, ABA must be involved in causing stomatal closure.

## 4. Materials and Methods

**Plant growth and transpiration measurement.** Barley seeds (*Hordeum vulgare* L. cv. Prairie) were germinated on tap water under ambient temperature. On the third day, germinated seedlings were transferred to 10% Hoagland–Arnon (H-A) solution and were grown at 24 ± 3 °C, 16 h light period, and 400 µmol m^2^ s^−1^ PAR. Seven days after the start of germination, intact plants were placed onto beakers with 50 mL of either 10% H-A solution (10 plants per beaker) (control) or a solution containing 100 mM NaCl or ABA (3 mg L^−1^) in addition for transpiration measurements in short-term experiments (several hours). Then, shoots were excised for measurement of hydraulic conductance and pieces of leaves were fixed for immunolocalization.

**Osmotic potential of xylem sap** was measured as described [22]. To collect xylem sap, barley seedlings were cut under water and the shoots reconnected to the roots with fine silicon tubing [23]. Comparing the transpiration rate of intact plants with cut and rejoined plants showed that this procedure did not significantly affect transpiration within 10 min, when water inside the tubing was replaced through xylem sap. After 10 min of collecting root exudate, the collection tubes were disconnected and the osmotic potential of root exudate collected from 10 plants was measured with a freezing point depression osmometer (Osmomat 030, Gonotech, Berlin, Germany). It was found to be −0.17 and −0.27 MPa for plants exposed to NaCl-free and NaCl solutions, respectively.

**Measurement of leaf hydraulic conductance.** Twenty minutes after addition of NaCl, leaves of control plants were dipped into solution simulating the osmotic potential of their xylem exudate (250% H-A solution having osmotic potential of −0.17 MPa). Leaves were cut in the solution and left with their bases inside it. The leaves of the salt-treated plants were cut off and left in a solution that mimicked the osmotic potential of the xylem exudate of these plants, which corresponded to 250% H-A solution + 20 mM NaCl having an osmotic potential of −0.27 MPa. Within 20 minutes after placing the leaves in these solutions, leaf transpiration was measured every 10 minutes as the loss of the weight by the beakers with the plants. The hydraulic conductance of the leaf (L), by analogy with Ohm’s law, was calculated using the equation: L = T/(Ψs − Ψl), where T is leaf transpiration rate (mg H_2_O plant^−1^ h^−1^), Ψs is the water potential of the nutrient solution, and Ψl is the leaf water potential, respectively (measured with a psychrometer) (modification of the method [24]).

**Cell pressure probe analyses.** Turgor, halftime of water exchange (T1/2), cell elastic modulus (e), and LpCell were determined for the ridge cells overlying large lateral veins through the cell pressure probe technique as described previously (e.g., [25,26,27]). Cell osmotic pressure, which is required for calculation of Lp, was determined through picolitre osmometry of sap extracted from cells [25], and the dimension of cells (volume, surface area) was determined through free-hand cross sections assuming that cells were shaped like cylinders. 

**ABA immunoassay.** For ABA assays, collected xylem sap was diluted with distilled water, acidified with HCl to pH 2.5, and partitioned twice with peroxide-free diethyl ether (the ratio of organic to aqueous phases was 1:3). Subsequently, hormones were transferred from the organic phase into 1% sodium bicarbonate (pH 7–8; the ratio of organic to aqueous phases was 3:1), re-extracted with diethyl ether, methylated with diazomethane, and immunoassayed using antibodies to ABA [10].

**Immunohistochemical localization of AQPs and ABA.** Immunolocalization was carried out on sections of the middle part of a mature leaf. To prevent ABA washing out from tissues during fixation and dehydration, leaf sections were fixed in 4% 1-ethyl-3-(3- dimethylaminopropyl) carbodiimide (Sigma, St. Louis, MO, USA) for 4 h (under vacuum during the first 30 min of fixation) and then with 4% formaldehyde overnight as described earlier [10]. In this process, ABA carboxylic groups are linked with protein amino groups. Following fixation in formaldehyde, leaf segments were dehydrated in ethanol solutions of increasing grades (up to 96%). Segments were then embedded in methacrylate resin (JB-4, Electron Microscopy Sciences, Hatfield, PA, USA) as recommended by the manufacturer. Histological sections were cut with a rotary microtome (HM 325, MICROM Laborgeräte, Berlin, Germany) and placed on slides. Sections were treated for 30 min with 0.1 M Na-phosphate buffer (PB) (pH 7.3) containing 0.2% gelatin and 0.05% Tween 20 (PGT), washed with distilled water and incubated for 2 h in a moist chamber at room temperature with immune rabbit anti-ABA or anti-HvPIP2;2 sera (20 µL) diluted with PGT at a ratio of 1:50 or 1:200, respectively. To check specificity of immunostaining, some sections were treated with nonimmune serum at similar dilution. To visualize antibodies bound to either ABA or AQPs, sections were treated for 1 h in a moist chamber with the second goat antibodies raised against rabbit immunoglobulin labeled with colloidal gold (1:40 in PGT; Aurion, Hatfield, PA, USA). After three washes with PB, samples were postfixed in 2% glutaraldehyde in PB for 5 min. The sections were then washed with distilled water and incubated with silver enhancer (Aurion) for 30 min. Excess silver was removed with distilled water, and sections were examined under a light microscope (Carl Zeiss, Jena, Germany) equipped with an AxioCam MRc5 digital camera (Carl Zeiss). The intensity of immunostaining of plasma membrane AQPs was semiquantified as described [10] from 8-bit grayscale images using ImageJ software (1.48, National Institutes of Health, Bethesda, MD, USA). Images were taken from 10 to 20 independent sections per variant of treatment. The intensity of staining was expressed in arbitrary units, with maximal staining taken as 100% and minimal as 0%. 

**Lignin staining with berberine hemisulfate.** Leaf pieces of barley plants were fixed in a mixture (7/7/100) of formalin, acetic acid, and ethanol 70%, dehydrated in an ethanol series, cleared with xylene, and embedded in paraffin. Sections were cut on a microtome and, after a dewaxing step, were stained using an aqueous solution of berberine hemisulfate (0.1% *w*/*v*) for 1 h and then rinsed 2 times with distilled water. To enhance the fluorescence intensity, sections were additionally stained for 15 min with toluidine blue (0.05% *w*/*v*) in 0.1 M PB (pH 5.6), rinsed 2 times with distilled water, mounted in a 0.1% FeCl_3_/50% glycerol mixture, and covered with a cover slip. Sections were excited with a 488 nm solid-state laser using a confocal laser scanning confocal microscope Olympus FluoView FV3000 (Olympus, Japan).

**Statistics**. The data were processed using the Statistica version 10 software (Statsoft, Moscow, Russia). In figures and tables, data are presented as mean ± standard error. Number of replications is provided in the figure and table legends. The significance of differences was assessed by ANOVA followed by Duncan’s test (*p* < 0.05).

## Figures and Tables

**Figure 1 ijms-23-14282-f001:**
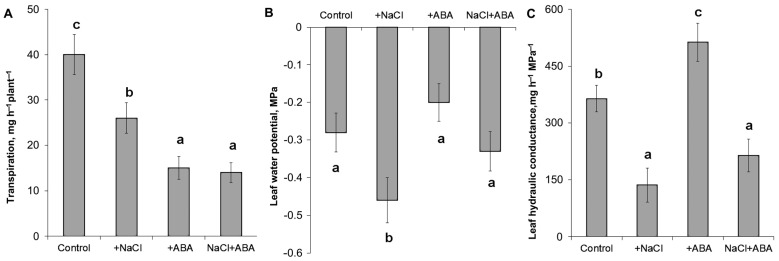
Effects of salinity and ABA treatment on transpiration (**A**), leaf water potential (**B**), and hydraulic conductance (**C**) of detached barley plant leaves. Statistically different values (n = 10) are labeled with different letters (Duncan test, *p* < 0.05).

**Figure 2 ijms-23-14282-f002:**
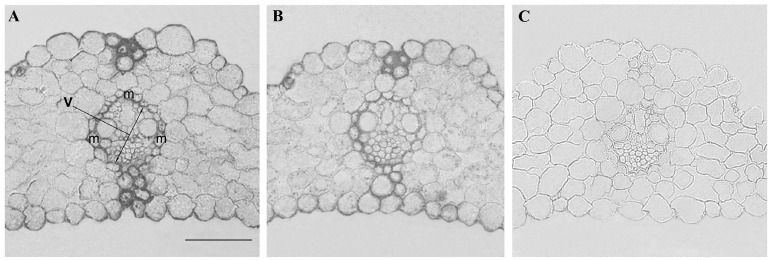
Immunohistochemical localization of HvPIP2;2 in mature leaf of 7-day-old barley of the control (**A**) and salt-treated plants (**B**) (1 h after addition of sodium chloride to the nutrient medium). Staining with nonimmune serum is also presented (**C**) (V—vasculature, m—mestome). Scale 100 µm.

**Figure 3 ijms-23-14282-f003:**
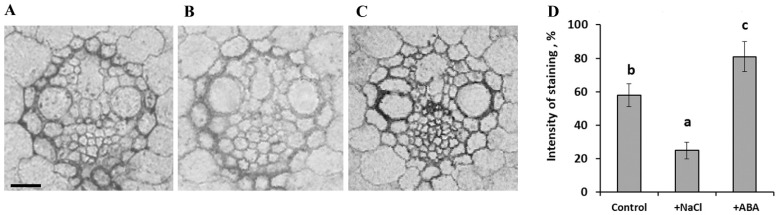
Immunohistochemical localization of HvPIP2;2 in the vasculature of leaves of the control (**A**), salt- (**B**), and ABA-treated (**C**) barley plants and semiquantitative estimation of the staining for HvPIP2;2 of vasculature cells (**D**). Statistically different values (n = 20) are labeled with different letters (Duncan test, *p* < 0.05). Scale 25 µm.

**Figure 4 ijms-23-14282-f004:**
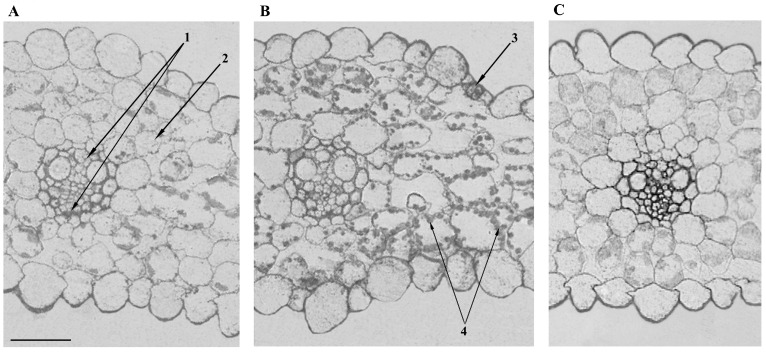
Immunohistochemical localization of ABA in the leaves of the control (**A**), salt- (**B**), and ABA-treated (**C**) barley plants. 1—vasculature, 2—mesophyll, 3—stomata, 4—chlorophyll. Scale 100 µm.

**Figure 5 ijms-23-14282-f005:**
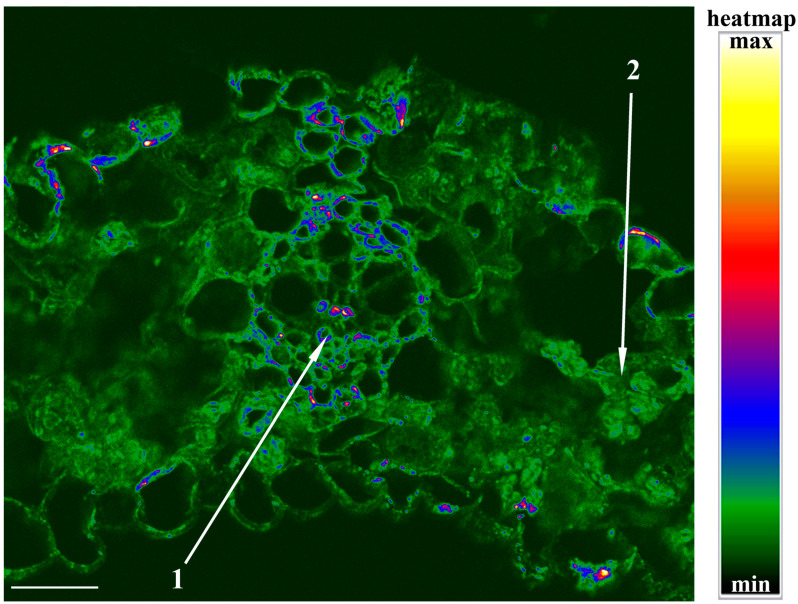
Localization of lignin in berberine-stained leaf cross sections of barley plants: 1—vasculature, 2—mesophyll. Heatmap represents intensity of fluorescence corresponding to lignin depicted by color. Scale 100 µm.

**Table 1 ijms-23-14282-t001:** Turgor and hydraulic conductivity (Lp) of ridge cells overlying large lateral veins.

	Turgor, bar	Lp, m s^−1^ MPa^−1^
control	11.98	7.53 × 10^−7^
salt treatment	8.77	7.32 × 10^−7^

## Data Availability

Data are contained within the article.

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
