# Peer review of "Effect of Salinity on Stomatal Conductance, Leaf Hydraulic Conductance, HvPIP2 Aquaporin, and Abscisic Acid Abundance in Barley Leaf Cells"

_ijms, 2022, doi:10.3390/ijms232214282_

Round 1

Reviewer 1 Report

I get your communication (Effect of salinity on stomatal conductance, leaf hydraulic conductance, HvPIP2 aquaporin and abscisic acid abundance in barley leaf cells) to review. In general, the manuscript represents a very big piece of information in a logical presentation. Therefore, it might be conditionally accepted subject to minor revision. Authors have to improve their manuscripts with many non-clear meanings, inaccuracies and inconsistencies, and the authors need to address the following issues before it can be accepted for publication.

1.     The introduction is short but has no clear thread.

2.     General note: the figures in this section are quite low resolution and difficult to make out. Higher-resolution versions will be needed for publication, for example, in Figures 2A, 2B; Figure 3A, 3B, 3C; Figure 4; and Figure 5.

3.     I would like the authors to provide the methodology and results of the number of replications wherever possible. The same applies to the statistical significance of the results. Please describe statistical methods used in the work in materials and methods.

4.     The values on the Y-axis of Figure B (such as -0,1 should be changed to -0.1) should be labeled in a standardized way.

5.     Line 193 (3 mg l-1) was changed to (3 mg·L-1). Please use international units in the article.

6.     Authors not follow the journal and references format, hence, check carefully the journal format and references.

Once the above concerns are fully addressed, the manuscript could be accepted for publication in this journal.

Author Response

We are most grateful to the reviewer for useful comment, which were carefully followed by us

  1. The introduction is short but has no clear thread.

Response: In order to follow clear thread, we introduced several sentences, which hopefully link several tasks fulfilled in the present research. The modified Introduction is as follows:

Balancing water losses and uptake by plants enables maintenance of hydraulic homeostasis. Stomata movement is an important mechanism preventing potentially harmful fluctuations in transpiration-induced changes in leaf water potential and turgor. Under conditions of lowered availability of water, stomatal closure reduces transpiration, which helps to maintain the hydration of tissues. The main attention of researchers has been addressed to the active uptake and efflux of osmotically-active solutes into and out of stomatal guard cells as the mechanism controlling stomatal movement [1]. However, it was shown that stomatal closure may be controlled by a decline in water flow into the leaves due to reduced ability of leaf tissues to conduct water [2]. Nevertheless, although a number of works show a functional relationship between the hydraulic and stomatal conductances of the leaf [3], the mechanisms underlying this relationship remain insufficiently studied. Water conductance by plant tissues depends on the presence and activity of water channel aquaporins in the membranes of cells [4]. The aim of the present work was to test for any involvement of aquaporins (AQPs) in stomatal closure under salinity. Previously we have shown radip stomatal closure after addition of sodium chloride to the nutrient medium of barley plants [5], associated with the fast accumulation of abscisic acid (ABA) [5], the hormone known to be involved in the control of stomatal closure [6]. ABA is also known to influence hydraulic conductance by increasing activity of AQPs [4]. Increased abundance of AQPs has been detected in barley roots treated with exogenous ABA [7]. However relation between stomatal conductance and shoot hydraulic conductivity and their possible relationship with the changes in ABA concentration were not considered. In the present work, the changes in the level of aquaporins in the leaf cells were detected with the help of immunohistochemical technique using antibodies against AQPs as described previously [7]. In parallel, leaf sections were stained for ABA. Changes in ABA content in the leaves were related to the influx of the hormone from the roots to reveal the role of ABA coming from the roots in the regulation of stomatal and hydraulic conductance in leaves. Furthermore, effects of salinity were compared to those of exogenously applied ABA on leaf AQPs levels and stomatal and leaf hydraulic conductance of barley plants.

  1. General note: the figures in this section are quite low resolution and difficult to make out. Higher-resolution versions will be needed for publication, for example, in Figures 2A, 2B; Figure 3A, 3B, 3C; Figure 4; and Figure 5.

Response: Figures with higher resolution are provided as tif files of supplementary material

3. I would like the authors to provide the methodology and results of the number of replications wherever possible. The same applies to the statistical significance of the results. Please describe statistical methods used in the work in materials and methods.

Response: We are sorry that due to the absence of the title of Statistic section in M & M, which was the very end of the article, it was unnoticed by the respected reviewer. We rectified this by introducing the section title in bold letters (Statistics). We also added that “Number of replications is provided in the figure and table legends” and checked mentioning number of replicates in the legends. n=10 is mentioned in Figure 1 and n=20  - in Figure 3D

4. The values on the Y-axis of Figure B (such as -0,1 should be changed to -0.1) should be labeled in a standardized way.

Response: commas were changed for points.

5.Line 193 (3 mg l-1) was changed to (3 mg·L-1). Please use international units in the article.

Response:corrected

6. Authors not follow the journal and references format, hence, check carefully the journal format and references.

Response: We used template provided by the journal and checked correctness of presenting the reference list

Reviewer 2 Report

Manuscript Review: „Effect of salinity on stomatal conductance, leaf hydraulic conductance, HvPIP2 aquaporin and abscisic acid abundance in barley leaf cells” written by Guzel Sharipova et al.

The topic is very interesting. The obtained results and conclusions significantly contribute to the development of knowledge about the role of aquaporins and abscisic acid in plants in salt stress conditions.

However, the manuscript in its present form is unpublishable. In my opinion, it requires some corrections.

1)     Introduction - too little information, laconic.

2)     The results are not precisely described. Only some parameters are described. The description for Figure 1C is incorrect.

3)     Figure 2, 3 and 4 with immunolocation is not clear, it is not visible where the antibodies have attached.

4)     Many of the results are recorded - "data not shown" - maybe it is worth showing them in supplements, or posting them in a repository and linking to it.

5)     For what purpose are the berberine experiments done?

Author Response

We are most grateful to the reviewer for useful comments, which were carefully followed by us.

  • Introduction - too little information, laconic.

Response; introduction was extended. It is as follow:

Balancing water losses and uptake by plants enables maintenance of hydraulic homeostasis. Stomata movement is an important mechanism preventing potentially harmful fluctuations in transpiration-induced changes in leaf water potential and turgor. Under conditions of lowered availability of water, stomatal closure reduces transpiration, which helps to maintain the hydration of tissues. The main attention of researchers has been addressed to the active uptake and efflux of osmotically-active solutes into and out of stomatal guard cells as the mechanism controlling stomatal movement [1]. However, it was shown that stomatal closure may be controlled by a decline in water flow into the leaves due to reduced ability of leaf tissues to conduct water [2]. Nevertheless, although a number of works show a functional relationship between the hydraulic and stomatal conductances of the leaf [3], the mechanisms underlying this relationship remain insufficiently studied. Water conductance by plant tissues depends on the presence and activity of water channel aquaporins in the membranes of cells [4]. The aim of the present work was to test for any involvement of aquaporins (AQPs) in stomatal closure under salinity. Previously we have shown radip stomatal closure after addition of sodium chloride to the nutrient medium of barley plants [5], associated with the fast accumulation of abscisic acid (ABA) [5], the hormone known to be involved in the control of stomatal closure [6]. ABA is also known to influence hydraulic conductance by increasing activity of AQPs [4]. Increased abundance of AQPs has been detected in barley roots treated with exogenous ABA [7]. However relation between stomatal conductance and shoot hydraulic conductivity and their possible relationship with the changes in ABA concentration were not considered. In the present work, the changes in the level of aquaporins in the leaf cells were detected with the help of immunohistochemical technique using antibodies against AQPs as described previously [7]. In parallel, leaf sections were stained for ABA. Changes in ABA content in the leaves were related to the influx of the hormone from the roots to reveal the role of ABA coming from the roots in the regulation of stomatal and hydraulic conductance in leaves. Furthermore, effects of salinity were compared to those of exogenously applied ABA on leaf AQPs levels and stomatal and leaf hydraulic conductance of barley plants.

  • The results are not precisely described. Only some parameters are described. The description for Figure 1C is incorrect.

Response: we are sorry for unclear description of Figure 1C. Thanks to the reviewer’s comment we noticed and rectified it by adding to the sentences that fold increase was compared to salt-stressed plants untreated with ABA: “Addition of ABA to salt-treated plants resulted in about 1.5 times increase in hydraulic conductance compared to the plants untreated with ABA and grown under salinity”.

We also added more description of the data. Thus we specified similarity in the changes in PIP abundance and hydraulic conductance by adding that “Changes in PIP2;2 abundance were in agreement with those in hydraulic conductance induced by NaCl and ABA-treatment (Figure 1C): a decline in both PIP2;2 abundance and hydraulic conductance in the case of salt-treated plants and increase in both characteristics in ABA treated plants compared to the control”

  • Figure 2, 3 and 4 with immunolocation is not clear, it is not visible where the antibodies have attached.

Response: To clarify specificity of binding of antibodies we added a figure showing staining of sections with unspecific serum. It shows very weak (almost absence) of staining. So we may believe that when specific antibodies are used, blackening corresponds to the presence of AQPs. To clarify presentation of the data, we added indication of mestome sheath, chloroplasts and some other structures. Figures with better resolution are now provided as supplementary data.

  • Many of the results are recorded - "data not shown" - maybe it is worth showing them in supplements, or posting them in a repository and linking to it.

Response: In accordance with this comment we added effect of salinity on transpiration of intact plants as Suppl. Figure 1.

Another case of mentioning “data not shown” concerns effect of salinity of concentration of ABA in xylem cell. Actually only two ciphers are meant and we added then in the brackets (21 ± 2 and 20 ± 3 ng ml-1 in the xylem of control and salt-treated plants, respectively). But thanks to the reviewer’s comment we realized that method of ABA assay in xylem sap was absent in the M & M section and we added it.

  • For what purpose are the berberine experiments done?

Importance of using berberine for detecting apoplast barrier was justified in the Discussion, where is it said that “The present experiments confirmed lignification of cells of mestome sheath (Figure 5), and formation of barriers for water movement is known to increase the importance of AQPs for the control of hydraulic conductivity, as AQPs allow water to move from cell to cell rather than along the apoplast, where barriers such as lignin are deposited. The importance of the decline in PIP2;2 abundance in mestome sheath of salt-stressed plants was confirmed by its correlation with the salt-induced decline in leaf hydraulic conductance.

However according to the remark of the reviewer we added a sentence justifying the use of berberine staining: “To detect formation of apoplast barriers we used barbering staining of lignin and suberin. Their staining showed increased fluorescence in periphery of cells of vasculature (Figure 5) coinciding with increased staining for PIP2;2 (Figure 2).”